# Prevalence of SARS-CoV-2 Variants of Concern and Variants of Interest in COVID-19 Breakthrough Infections in a Hospital in Monterrey, Mexico

**DOI:** 10.3390/v14010154

**Published:** 2022-01-14

**Authors:** Kame A. Galán-Huerta, Samantha Flores-Treviño, Daniel Salas-Treviño, Paola Bocanegra-Ibarias, Ana M. Rivas-Estilla, Eduardo Pérez-Alba, Sonia A. Lozano-Sepúlveda, Daniel Arellanos-Soto, Adrián Camacho-Ortiz

**Affiliations:** 1Center for Investigation and Innovation in Medical Virology, Department of Biochemistry and Molecular Medicine, School of Medicine, Universidad Autónoma de Nuevo León, Monterrey 64460, Mexico; kame.galanhr@uanl.edu.mx (K.A.G.-H.); amrivas1@yahoo.ca (A.M.R.-E.); lozano_sonia@hotmail.com (S.A.L.-S.); d_arellanos_s@yahoo.com (D.A.-S.); 2Department of Infectious Diseases, School of Medicine, University Hospital “Dr. José Eleuterio González”, Universidad Autónoma de Nuevo León, Monterrey 64460, Mexico; samflorest@gmail.com (S.F.-T.); danielsalast91@gmail.com (D.S.-T.); paola.bocanegraib@gmail.com (P.B.-I.); md.eduardo.perez@gmail.com (E.P.-A.)

**Keywords:** SARS-CoV-2, COVID-19, Latin America, Mexico, COVID-19 vaccines, surveillance, variant of concern, Delta, Mu, CanSino

## Abstract

SARS-CoV-2 variants of concern (VOCs) or of interest (VOIs) causing vaccine breakthrough infections pose an increased risk to worldwide public health. An observational case-control study was performed of SARS-CoV-2 vaccine breakthrough infections in hospitalized or ambulatory patients in Monterrey, Mexico, from April through August 2021. Vaccination breakthrough was defined as a SARS-CoV-2 infection that occurred any time after 7 days of inoculation with partial (e.g., first dose of two-dose vaccines) or complete immunization (e.g., second dose of two-dose vaccines or single-dose vaccine, accordingly). Case group patients (*n* = 53) had partial or complete vaccination schemes with CanSino (45%), Sinovac (19%), Pfizer/BioNTech (15%), and AstraZeneca/Oxford (15%). CanSino was administered most frequently in ambulatory patients (*p* < 0.01). The control group (*n* = 19) received no COVID-19 vaccines. Among SARS-CoV-2 variants detected by whole-genome sequencing, VOC Delta B.1.617.2 predominated in vaccinated ambulatory patients (*p* < 0.01) and AY.4 in hospitalized patients (*p* = 0.04); VOI Mu B.1.621 was detected in four (7.55%) vaccinated patients. SARS-CoV-2 breakthrough infections in our hospital occurred mostly in patients vaccinated with CanSino due to the higher prevalence of CanSino vaccine administration in our population. These patients developed mild COVID-19 symptoms not requiring hospitalization. The significance of this study lies on the detection of SARS-CoV-2 variants compromising the efficacy of local immunization therapies in Monterrey, Mexico.

## 1. Introduction

The coronavirus disease 2019 (COVID-19) emerged in December 2019 and was later declared by the World Health Organization as a pandemic [1]. COVID-19 is caused by the severe acute respiratory syndrome coronavirus 2 (SARS-CoV-2), and there are currently more than 200 million cases and over 4 million deaths reported worldwide. In Mexico, 3 million cases and up to 250,000 deaths have been reported [2].

The rapid development of efficient and safe vaccines against COVID-19 undertook countless efforts in the year 2020 [3]. Nowadays, up to 4.5 billion vaccine doses have been already administered worldwide [2]. Mexico started administering the BNT162b2 (Pfizer/BioNTech) vaccine to its health personnel in late December of 2020, soon after it was declared for emergency use by the WHO. BNT162b2 is a lipid nanoparticle-encapsulated nucleoside-modified RNA vaccine that encodes the full-length surface spike (S) glycoprotein of SARS-CoV-2 [4,5]. Since then, Ad5-nCoV (CanSino Biologics Inc.) and AZD1222 (AstraZeneca/University of Oxford), both nonreplicating recombinant adenovirus vectors that express the S protein [4,5,6,7], and the (Vero cell) inactivated whole-virus CoronaVac (Sinovac Life Sciences) vaccine [8] have also been administered in the country. Nevertheless, governmental vaccine acquisition has been laborious, and reduced widespread vaccine availability and delayed vaccine uptake have caused only 85.3 million vaccine doses to be administered in Mexico [9].

Moreover, SARS-CoV-2 variants have emerged in several places of the world carrying several mutations in the S protein [10]. These variants are classified by the WHO as variants of concern (VOCs) and variants of interests (VOIs). Variants of Concern can either have increased transmissibility, virulence, severe disease presentation, reduced diagnostic detection, vaccine effectiveness, or treatment effectiveness. Current VOCs are Alpha (B.1.1.7), Beta (B.1.351), Gamma (P.1), Delta (B.1.617.2), and Omicron (B.1.1.529). The VOIs have genetic changes that affect transmissibility, disease presentation, and immune and therapeutic effectiveness, in addition to significant community transmission posing a risk to global public health. The current VOIs designated by WHO are Lambda (C.37) and Mu (B.1.621) [11]. Such variants can confer superior viral replication or transmission and compromise the efficacy of current immunization therapies [7]. Thus, the transmission of these variants is worrisome as they pose an increased risk to worldwide public health [7,10].

The aim of this study was to describe the prevalence of SARS-CoV-2 VOCs or VOIs in patients with partial or complete vaccination schemes in Monterrey, Mexico.

## 2. Materials and Methods

### 2.1. Setting

The University Hospital “Dr. José Eleuterio Gonzalez” located in Monterrey, Mexico, is a multi-building hospital and medical school complex that comprises two separate hospitals. The main building is a 600-bed hospital designated for the treatment of non-COVID-19 cases. It has an average of 25,000 admissions per year. The second building is an 85-bed hospital that was designated and first used for the diagnosis and treatment of suspected or confirmed COVID-19 cases. It has an outpatient clinic, an emergency room, hospitalization wards, an intensive care unit (ICU), a step-down unit, and two operating rooms.

### 2.2. Selection Criteria

The design is an observational case-control study of SARS-CoV-2 vaccine breakthrough infections in hospitalized patients in our unit or who attended the clinic from April through August 2021. Individuals with COVID-19 confirmed by detection of SARS-CoV-2 by real-time reverse transcription PCR (rRT-PCR) were included. The case group was composed of individuals aged >18 years with a positive rRT-PCR for SARS-CoV-2 and Ct ≤ 30, either hospitalized (i.e., with supplemental oxygen requirement and abnormal chest imaging) or ambulatory patients (i.e., with mild symptoms and without oxygen requirement). The infection occurred any time after 7 days of inoculation with the first dose of Pfizer/BioNTech, AstraZeneca/Oxford, or CoronaVac, or with the single dose CanSino, according to patients’ records. The control group consisted of randomly selected SARS-CoV-2-positive cases that were also hospitalized occurred in the same period as the breakthrough infections.

### 2.3. Whole-Genome Sequencing and Lineage Determination

RNA was extracted from 200 µL of nasopharyngeal swab specimens transported in UTM viral transport medium (COPAN Diagnostics Inc., Murrieta, CA, USA) using a STARMag Universal Cartridge Kit (Seegene Technologies, Seoul, South Korea) on a Nimbus extraction workstation (Seegene Technologies) following the manufacturer’s instructions. SARS-CoV-2 detection was performed with a Smart Detect SARS-CoV-2 rRT-PCR Kit (InBios International, Inc., Seattle, WA, USA) on a CFX96 thermal cycler (Bio-Rad Laboratories, Hercules, CA, USA).

The improved ARTIC multiplex PCR method for SARS-CoV-2 genome sequencing was performed using nanopore technology [12]. Viral RNA (8 µL) was converted to cDNA with LunaScript RT (New England Biolabs, Ipswich, MA, USA) and later used in a multiplex PCR using Q5 Hot Start HF polymerase (New England Biolabs). Ends were repaired with NEBNext Ultra II end repair/dA-tailing module (New England Biolabs), and native barcodes (Oxford Nanopore Technologies, Oxford, United Kingdom) were added with Blunt/TA ligase (New England Biolabs). After washing with AMPure XP beads (Beckman Coulter, Brena, CA, USA), adapters (Oxford Nanopore Technologies, Oxford, United Kingdom) were added with the NEBNext quick ligation module (New England Biolabs). The library was sequenced on an R9.4.1 flow cell (Oxford Nanopore Technologies, Oxford, United Kingdom) for 12 h. The sequences were assembled using the ARTIC network bioinformatics pipeline via nanopolish and the reference sequence MN908947.3 to obtain the consensus sequences. SARS-CoV-2 lineages were obtained using the Pangolin COVID-19 Lineage Assigner web application (https://pangolin.cog-uk.io/, accessed on 6 September 2021) [13]. Sequences with more than 10% of Ns were discarded from the study.

### 2.4. Phylogenetic Analyses

Global reference sequences were selected from the global dataset (*n* = 3493) obtained from Nextstrain [14] and were retrieved from the Global Initiative for Sharing All Influenza Data (GISAID) database. Sequence quality was determined with Nextclade web tool, and 58 low-quality sequences were removed. The final sequence dataset was composed of 3499 sequences (3435 from the global dataset and 64 generated from this study). Afterward, Nextstrain was used to analyze these sequences. Phylogenetic trees were visualized with ggtree [15]. The authors of sequences used are listed in Appendix A. The sequences generated were deposited in GISAID (EPI_ISL_4889733—EPI_ISL_4889791 and EPI_ISL_4926951—EPI_ISL_4926955).

### 2.5. Statistical Analyses

Continuous variables, such as age and Ct values, are expressed as means. The dispersion of these variables is expressed in interquartile values. The statistical mean differences were evaluated using the one-way ANOVA test. The normal distribution of these variables was verified using the Kolmogorov–Smirnoff test. Contingency tables using the χ^2^ and Fisher’s exact statistical tests (2 × 2 tables) were used to evaluate discontinuous variables. In all tests, a *p*-value < 0.05 was considered significant.

## 3. Results

### 3.1. Breakthrough Infections in Monterrey, Mexico

A total of 53 cases of COVID-19 vaccine breakthrough infections occurred in our hospital from 19 April 2021 to 11 August 2021 (Appendix A). The definition for vaccination breakthrough in this study was any SARS-CoV-2 infection that occurred any time after 7 days of inoculation with partial (e.g., first dose of two-dose vaccines) or complete immunization (e.g., second dose of two-dose vaccines or one single-dose vaccine, accordingly). These patients were vaccinated with CanSino (*n* = 24, 45.28%) between March and May, Sinovac (*n* = 10, 18.87%) between April and June, and equally with Pfizer/BioNTech (*n* = 8, 15.09%) between January and August and AstraZeneca/Oxford (*n* = 8, 15.09%) between April and May. The comparison of vaccinated ambulatory patients (*n* = 30, 56.60%) and those who required hospitalization (*n* = 23, 43.40%) showed that the single-dose CanSino was administered most frequently in ambulatory patients (*p* < 0.01). Furthermore, almost all ambulatory patients received complete vaccination schemes regardless of vaccine type before they developed COVID-19-related symptoms, unlike hospitalized patients (*p* = 0.03).

In the study, 19 unvaccinated hospitalized patients (control group) were also included in the same period as the case-control group. Hospitalized vaccinated patients had a higher median age (*p* < 0.01) and comorbidities frequency, such as hypertension (*n* = 11, 47.82%, *p* = 0.02) and diabetes mellitus type 2 (*n* = 13, 56.52%, *p* < 0.01), compared to ambulatory or unvaccinated hospitalized patients. No differences were found between vaccinated (hospitalized and ambulatory) and unvaccinated patients regarding oxygen requirement, clinical intervention, or hospital treatment (Table 1).

### 3.2. SARS-CoV-2 Lineages and Variants

At the time of the study, the VOCs were Alpha (B.1.1.7), Beta (B.1.351, B.1.351.2, and B.1.351.3), Gamma (P.1, P.1.1, and P.1.2), and Delta (B.1.617.2, AY.1, AY.2, AY.3, and AY.4). The VOIs were Eta (B.1.525), Iota (B.1.526), Kappa (B.1.617.1), Lambda (C.37), and Mu (B.1.621) [11]. Thus, the comparison studies were performed using this classification.

SARS-CoV-2 variants detected in vaccine breakthrough infections were Delta (*n* = 36, 67.92%), Gamma (*n* = 4, 7.55%), Mu (*n* = 4, 7.55%), and Alpha (*n* = 3, 5.66%), which spanned through the phylogenetic tree shown in Figure 1A. In contrast, Delta (*n* = 17, 89.5%) and Gamma (*n* = 2, 10.5%) were detected in non-vaccinated hospitalized patients. In breakthrough infections, the four Gamma viruses detected infected patients regardless of their status (ambulatory or hospitalized, vaccinated or non-vaccinated) in our population, and were related to independent introductions (Figure 1B). Mexican Gamma isolates spanned throughout the Gamma clade. Viruses isolated in this study did not group directly with other Mexican isolates. In our study, only three Alpha viruses were detected: two in ambulatory patients and the other in a vaccinated hospitalized patient. However, only one Alpha virus could be analyzed due to sequence quality (Figure 1C). Mexican Alpha isolates distributed similarly to Gamma isolates, thus confirming co-circulation of these VOCs. Four Mu viruses were detected: two in vaccinated hospitalized patients and two in ambulatory patients. None were detected in non-vaccinated patients. Three of the Mu viruses were clustered in the same clade (Figure 1D). At the time of analysis, there were only two more Mexican Mu isolates. One isolate grouped with one of the isolated reported here, suggesting a transmission event. The other isolate was unrelated.

Different Delta strains were detected and distributed throughout the tree (Figure 2A). Delta sequences from Mexico grouped into specific clades, which suggested few introductions. Our sequenced viruses grouped into five clades (Figure 2). The first clade contained 20 samples from ambulatory, hospitalized, and non-vaccinated hospitalized patients (Figure 2B). Within this clade, 13 of the 20 isolated viruses were clustered into a single subclade, indicating a transmission chain. The second clade contained 11 samples from our study (Figure 2F), and the remaining three clades consisted of several viruses (Figure 2C–E). Delta isolated viruses were detected in several Mexican states, thus confirming the wide spread of this variant in Mexico. The frequency of VOCs was 81.13% (*n* = 43) in all vaccinated patients included in the study. An upward trend was observed in vaccinated patients who were infected by the Delta variant (*p* = 0.08), as well as in those who contracted an infection by VOCs (*p* = 0.05), compared to patients who were not vaccinated.

Detected SARS-CoV-2 lineages were highly diverse (Table 2). B.1.617.2 (Delta variant) predominated in vaccinated ambulatory patients (*n* = 15, 68.18%, *p* < 0.01) compared to hospitalized patients regardless of vaccine status. Instead, AY.4 (Delta variant) predominated in either vaccinated or non-vaccinated hospitalized patients compared to ambulatory patients (*p* = 0.04). The distribution of remaining lineages between vaccinated and unvaccinated patients showed no statistical difference, suggesting both groups were similarly affected by circulating SARS-CoV-2 variants.

## 4. Discussion

COVID-19 vaccination is one of the most effective methods to control the pandemic [16]. However, vaccinated individuals can still become infected and develop COVID-19 symptoms. In these breakthrough infections, SARS-CoV-2 RNA or antigen is detected ≥14 days after receiving recommended doses of a COVID-19 vaccine [17]. Severe COVID-19 disease can occur in 10% of individuals with breakthrough infections, in which older age is considered a risk factor [18]. Consequently, lower risk of severe disease is associated with vaccination. Some of the vaccines authorized for emergency use in Mexico (i.e., Pfizer/BioNTech and AstraZeneca/Oxford) offer variable protection levels against SARS-CoV-2 VOCs. After complete immunization, the effectiveness against symptomatic infection of Pfizer/BioNTech decreased from 93.7% against the Alpha variant to 88.0% with the Delta variant. Additionally, the effectiveness of AstraZeneca/Oxford declined from 74.5% against the Alpha variant to 67.0% with the Delta variant. Nevertheless, after only one dose, both vaccines were only 48.7% and 30.7% effective against the Alpha and Delta variants, respectively [19].

After mRNA vaccine administration such as Pfizer or Moderna (mRNA-1273, Moderna, Cambridge, MA, USA), breakthrough confirmed cases were reported worldwide, mainly caused by VOCs such as Alpha B.1.1.7 [20,21,22,23] and Beta B.1.351 [21,24], in Israel [20,21], Greece [22], Germany [24], and Belgium [23]. The P1 (Gamma VOC) variant was reported in patients with CoronaVac from Brazil [25]. In the USA, a predominance of B.1.526 (Iota VOI) or B.1.1.7 (Alpha VOC) occurred in breakthrough infections in New York [26,27], Massachusetts [28], and other parts of the country [29]. Delta variants such as B.1.617.2 and AY.3 were reported in Massachusetts [30], in addition to P.1 (Gamma), and B.1.351 (Beta) in Washington state [30]. In our study, widespread transmission of the Gamma variant was detected in Mexico, as up to 80% of the Gamma viruses were isolated in Mexico. One of the three Alpha viruses was clustered with viruses isolated in Austria and New Zealand.

The Mu (B.1.621) variant carries several known spike mutations, such as E484K, N501Y, and P681H; and some new mutations, such as R346K, Y144T, Y145S, and 146N. This variant was first detected in Colombia in January 2021, where its transmission rapidly increased [31], and it was later imported into Italy from a traveler coming from Colombia in April 2021 [32]. The Mu variant was recently designated as a VOI by the WHO on 30 August 2021. To date, the B.1.621 lineage is present in Colombia, the United States, Spain, the Netherlands, Denmark, Germany, Curacao, and Mexico. Particularly in Mexico, we detected four Mu viruses, three of which were grouped in the same clade, possibly indicating a single transmission event. The fourth Mu virus was grouped into another clade with a virus isolated in Puebla, Mexico. These results showed that the Mu variant was introduced into Mexico on different occasions. Mu viruses were also grouped with viruses isolated in Ecuador and Colombia.

The B.1.617.2 Delta variant was first detected in India, later becoming dominant in England [33]. This VOC contains mutations in the S viral protein, which is associated with increased viral transmission and infectivity, in addition to higher hospitalization risk [7]. Delta variants (B.1.617.2 and AY.4) were the cause of most of the breakthrough infections in the Monterrey metropolitan area (67.92%). Delta viruses were clustered into five clades, most of them with high frequency of viruses previously isolated from different Mexican states. The first clade contained 20 samples in addition to a high number of viruses isolated in Mexico (84%). Within this clade, 13 of the 20 isolated viruses were clustered into a single subclade, indicating a transmission chain. The second clade also contained a high number Mexican isolates (91%) and 11 samples from our study, which clustered with viruses isolated from different Mexican states. The remaining Delta viruses’ clades were also clustered with viruses isolated in Costa Rica, and Guatemala.

In Mexico, a remarkable decline in COVID-19 cases occurred in early spring 2021, due in part to governmental vaccination campaigns in populations at high risk (i.e., health personnel and elderly people) in late December 2020 and early February 2021 [34]. However, during May and June 2021, COVID-19 cases, hospitalizations, and deaths started to increase nationwide soon after the Delta variant was designated as a VOC by the WHO. The highest peak ever reported in Mexico since the pandemic officially started in the country (first government-confirmed COVID-19 case on 28 February 2020) occurred recently on 12 August 2021, with 24,975 new daily cases and 608 deaths [35]. Likewise, COVID-19 cases in EUA increased 300% during June and July 2021 caused by the Delta variant, which now has a prevalence of 83.4% [16,36].

The national response to the COVID-19 health emergency varied considerably within the country in the first months of the pandemic. Mexican states had widely different responses in terms of the implementation and timing of measure policies to mitigate the spread of COVID-19. However, the performance of the state of Nuevo Leon was one of the highest in the country [37], as the state government established policies to promote social distancing, such as the suspension of on-site classes and mass events, the reinforcement of economic restrictions, and the stay-at-home request, before federal measures were implemented. Since then, health policies have been highly dynamic with the subsequent lifting of social distancing measures and public space capacity limits that have contributed to a sustained virus transmission in the region [38].

Vaccination efforts should also be accelerated to reduce community transmission. As of 31 August 2021, the proportion of the total Mexican population fully vaccinated was 26.8%, and 38.4% in the population over 18 years of age [9]. The risk of infection, severe illness, and death in non-vaccinated people is highly significant [16] and vaccination refusal has also complicated the situation [17].

In the first stages of vaccine administration in Mexico, the administration of Pfizer/BioNTech vaccines was directed toward prioritized population groups (i.e., health personnel attending COVID-19 patients and elderly people), whereas the teaching staff was later vaccinated with CanSino with the hopeful aim of opening schools soon [34]. However, current trends indicate the level of SARS-CoV-2 community transmission remains high [39]. The increase in case detection rates implies that prevention strategies need to be strengthened [16]. COVID-19 incidence should be monitored in populations of high risk of exposure, such as students and teaching personnel, in addition to healthcare workers [16,40].

Our results showed a higher prevalence of CanSino vaccine administration in our population (45.28%) compared to other vaccine brands in the same period. Between April and May 2021, the national government implemented the administration of the CanSino vaccine exclusively in teaching staff, including the state of Nuevo Leon [41]. By the end of May, 87% of the teaching personnel in Mexico was fully vaccinated [42]. In addition, ours is a university teaching hospital, in which a great part of the teaching staff is attended.

The majority of patients vaccinated with CanSino who had breakthrough infections (66.66%) developed milder symptoms of COVID-19 that did not require hospitalization (*p* < 0.01). Although our results were not conclusive regarding other types of vaccines, Pfizer/BioNTech and AstraZeneca/Oxford were 88.0% and 67.0% effective in reducing symptomatic Delta infection, respectively [19]. In our study, patients who had a complete vaccination scheme were less likely to develop severe COVID-19 disease requiring hospitalization compared to those who received partial or incomplete immunization.

One limitation of this study is the inclusion of patients with partial vaccination who only had one of the two doses recommended for Pfizer/BioNTech, AstraZeneca/Oxford, or CoronaVac [6,16]. However, follow-up of instructions to complete vaccination schemes in people vaccinated with one dose is sometimes complicated. Another limitation is the small sample of patients included in the study. Better analyses could be performed if a higher number of samples can be studied.

In conclusion, SARS-CoV-2 breakthrough infections that occurred in our hospital were mostly in patients vaccinated with CanSino due to a higher prevalence of CanSino vaccine administration in our population compared to other vaccine brands in the same period. In our population, most of the patients vaccinated with CanSino who had breakthrough infections developed mild symptoms, and few patients required hospitalization. Patients who had a complete vaccination scheme were less likely to develop severe COVID-19 disease requiring hospitalization compared to those who received partial or incomplete immunization. Delta variants, mainly B.1.617.2 and AY.4, were the cause of most of the breakthrough infections. The recently designated Mu variant was detected in four of our vaccinated patients. The relatively modest number of sequenced samples poses another limitation for the extrapolation of data to other regions, although a strength of this group lies in the valuable data regarding infections in persons vaccinated with CanSino and Sinovac, for whom data are scant.

## Figures and Tables

**Figure 1 viruses-14-00154-f001:**
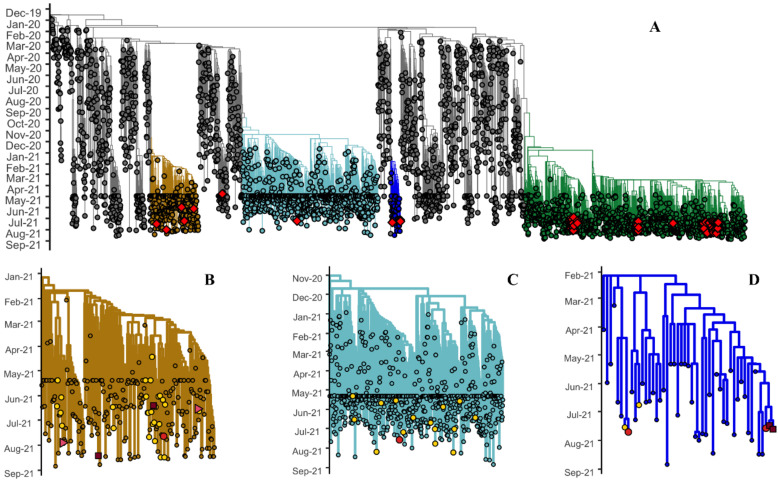
Phylogenetic relationship of SARS-CoV-2, 2019–2021. Global context tree highlighting the different variants: (**A**) Gamma (gold), Alpha (teal), Mu (blue), and Delta (green). Red diamonds indicate the viruses obtained in this study. The relationships of the different variants detected in the studied patients were magnified into (**B**) Gamma, (**C**) Alpha, and (**D**) Mu. Yellow circles indicate viruses isolated in Mexico. Red circles indicate ambulatory vaccinated patients. Red squares indicate hospitalized vaccinated patients. Pink triangles indicate hospitalized non-vaccinated patients. Sequences used in this analysis were downloaded from the Global Initiative for Sharing All Influenza Data (GISAID) database.

**Figure 2 viruses-14-00154-f002:**
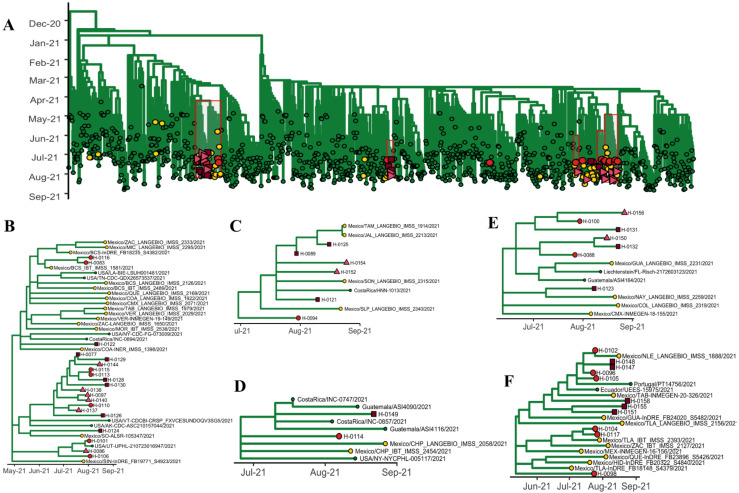
Delta variant phylogenetic relationship. Phylogenetic tree showing the analyzed Delta viruses and the distribution of the obtained sequences (**A**). Red boxes indicate posterior magnification of the analyzed clades, which are the following: the largest clade involving 20 of the generated sequences (**B**) and the minor clades in which the obtained sequences grouped with other Mexican isolates (**C**–**F**). Yellow circles indicate viruses isolated in Mexico. Red circles indicate ambulatory vaccinated patients. Red squares indicate hospitalized vaccinated patients. Pink triangles indicate hospitalized non-vaccinated patients. Sequences used in this analysis were downloaded from the Global Initiative for Sharing All Influenza Data (GISAID) database.

**Table 1 viruses-14-00154-t001:** Baseline data, comorbidities, vaccination status, and clinical intervention of patients included.

Characteristic ^1^	Vaccinated *n* (% or Range)		Unvaccinated *n* (% or Range)	*p*-Value ^2^
	Hospitalized (*n* = 23)	Ambulatory (*n* = 30)	Hospitalized (*n* = 19)	
Male	12 (52.17)	13 (43.33)	9 (47.37)	
Female	11 (47.83)	17 (56.67)	10 (52.63)	
Median age (IQR), years	59.70 (50–70)	42.80 (27.75–56.25)	48.47 (39–62)	**<0.01**
Ct (IQR)	19.58 (17.19–22.49)	18.81 (15.72–21.24)	19.00 (16.70–20.83)	0.72
Vaccine administered				
Astra Zeneca/Oxford	5 (21.73)	3 (10.00)	0 (0.00)	0.50
Pfizer/BioNTech	5 (21.73)	3 (10.00)	0 (0.00)	0.50
CanSino	4 (17.39)	20 (66.66)	0 (0.00)	**<0.01**
Sinovac	6 (26.08)	4 (13.33)	0 (0.00)	0.50
Unspecified	3 (13.04)	0 (0.00)	0 (0.00)	
Complete vaccination scheme	17 (73.91)	29 (96.66)	0 (0.00)	**0.03**
Supplementary oxygen at arrival time	14 (60.86)	0 (0.00)	11 (57.89)	
High-flow O_2_	14 (60.86)	0 (0.00)	6 (31.57)	0.07
Intubation	10 (43.47)	0 (0.00)	8 (42.10)	
ICU admission	1 (4.34)	0 (0.00)	0 (0.00)	
Comorbidities				
Hypertension	11 (47.82)	5 (16.66)	3 (15.78)	**0.09**
Diabetes mellitus type II	13 (56.52)	5 (16.66)	5 (26.31)	**<0.01**
Obesity	1 (4.34)	2 (6.66)	3 (15.78)	0.37
Smoking	4 (17.39)	1 (3.33)	2 (10.52)	0.23
Treatment				
Tocilizumab	1 (4.34)	0 (0.00)	0 (0.00)	
Baricitinib	16 (69.56)	0 (0.00)	9 (47.36)	0.21
Dexamethasone	19 (82.60)	0 (0.00)	12 (63.15)	0.27
Methylprednisolone	17 (73.91)	0 (0.00)	12 (63.15)	0.52
Mortality	4 (17.39)	0 (0.00)	3 (15.78)	

^1^ IQR: interquartile range; ICU: intensive care unit. ^2^ Significant *p*-values are denoted in bold letters.

**Table 2 viruses-14-00154-t002:** Descriptive data of SARS-CoV-2 variants and lineages detected in this study.

Variant or Lineage ^1^	Vaccinated *n* (% or Range)		Unvaccinated *n* (% or Range)	*p*-Value ^2^
	Hospitalized (*n* = 23)	Ambulatory (*n* = 30)	Hospitalized (*n* = 19)	
Alpha	1 (4.34)	2 (6.66)	0 (0.00)	0.52
B.1.1.7	1 (100)	2 (100)	0 (0.00)	
Gamma	2 (8.68)	2 (6.66)	2 (10.52)	0.89
P.1	2 (100)	1 (50.00)	2 (100)	
P.1.1	0 (0.00)	1 (50.00)	0 (0.00)	
Delta	14 (60.86)	22 (73.33)	17 (89.47)	0.11
AY.11	5 (35.71)	0 (0.00)	4 (23.52)	0.09
AY.3	3 (21.42)	6 (27.27)	2 (11.76)	0.49
AY.4	4 (28.57)	1 (4.54)	6 (35.29)	**0.04**
AY.5	0 (0.00)	0 (0.00)	2 (11.76)	
B.1.617.2	2 (14.28)	15 (68.18)	3 (17.64)	**<0.01**
Mu	2 (8.68)	2 (6.66)	0 (0.00)	0.96
B.1.621	2 (100)	2 (100)	0 (0.00)	
Other lineages	4 (17.39)	2 (6.66)	0 (0.00)	0.47
B.1	1 (25.00)	1 (50.00)	0 (0.00)	
B.1.1.519	1 (25.00)	0 (0.00)	0 (0.00)	
B.1.618	0 (0.00)	1 (50.00)	0 (0.00)	
B.1.628	1 (25.00)	0 (0.00)	0 (0.00)	
B.1.632	1 (25.00)	0 (0.00)	0 (0.00)	
VOCs	17 (73.91)	26 (86.66)	19 (100)	0.05
VOIs	2 (8.68)	2 (6.66)	0 (0.00)	0.44

^1^ VOCs: variants of concern; VOIs: variants of interest. ^2^ Significant *p* values are denoted in bold letters.

## Data Availability

The sequences generated were deposited in GISAID (EPI_ISL_4889733—EPI_ISL_4889791 and EPI_ISL_4926951—EPI_ISL_4926955).

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
