# Peer review of "Prevalence of SARS-CoV-2 Variants of Concern and Variants of Interest in COVID-19 Breakthrough Infections in a Hospital in Monterrey, Mexico"

_viruses, 2022, doi:10.3390/v14010154_

Round 1

Reviewer 1 Report

In this manuscript “Prevalence of SARS-CoV-2 Variants of Concern and Variants of 2 Interest in COVID-19 Breakthrough Infections in Mexico” by Kame A. Galán-Huerta and colleagues the authors describe their experience with vaccine breakthrough for several platform vaccines. The study took place between April 2021 through August 2021.  The patients were ambulatory or hospitalized. Four vaccines Sinovac, CanSino, AstraZeneca, or Pfizer were in use. Based on the presented data, CanSino vaccinated group had the highest rate of breakthrough.  Vaccine breakthrough for other three vaccines were very similar and were lower. The investigators performed whole genome sequencing to determine the circulating/breakthrough variant(s).  The authors report B.1.617.2 was the predominant in vaccinated ambulatory patients. Interestingly, AY.4 was the most predominant in hospitalized individuals. May be of bigger concern was the detection of the Mu (B.1.621) variant which carried many mutations in the spike and was first detected in Colombia in January of 2021 and then was exported to other countries.

Overall, this is a nicely put together manuscript and the data are important for public health management of the pandemic. I have a single concern about this manuscript. The authors do not mention when the vaccines were given and when the infections occurred in these patients.  This is important to assure the readers that CanSino vaccination occurred within the same time frame as other vaccines.  This concern should be addressed before the manuscript goes forward.

Author Response

Response to Reviewer 1 Comments

In this manuscript “Prevalence of SARS-CoV-2 Variants of Concern and Variants of 2 Interest in COVID-19 Breakthrough Infections in Mexico” by Kame A. Galán-Huerta and colleagues the authors describe their experience with vaccine breakthrough for several platform vaccines. The study took place between April 2021 through August 2021. The patients were ambulatory or hospitalized. Four vaccines Sinovac, CanSino, AstraZeneca, or Pfizer were in use. Based on the presented data, CanSino vaccinated group had the highest rate of breakthrough. Vaccine breakthrough for other three vaccines were very similar and were lower. The investigators performed whole genome sequencing to determine the circulating/breakthrough variant(s).  The authors report B.1.617.2 was the predominant in vaccinated ambulatory patients. Interestingly, AY.4 was the most predominant in hospitalized individuals. May be of bigger concern was the detection of the Mu (B.1.621) variant which carried many mutations in the spike and was first detected in Colombia in January of 2021 and then was exported to other countries. Overall, this is a nicely put together manuscript and the data are important for public health management of the pandemic.

Point 1: I have a single concern about this manuscript. The authors do not mention when the vaccines were given and when the infections occurred in these patients. This is important to assure the readers that CanSino vaccination occurred within the same time frame as other vaccines. This concern should be addressed before the manuscript goes forward.

Response 1: We thank the reviewer for their comments and observations. We have included this missing data in the Results section: “A total of 53 cases of COVID-19 vaccine breakthrough infections occurred in our hospital from April 19th, 2021, to August 11th, 2021. These patients were vaccinated with CanSino (n=24, 45.28%) between March and May, Sinovac (n=10, 18.87%) between April and June, and equally with Pfizer/BioNTech (n=8, 15.09%) between January and August, and AstraZeneca/Oxford (n=8, 15.09%) between April and May.”

Reviewer 2 Report

As new SARS-CoV 2 variants continue to emerge globally,  breakthrough infection by variants in vaccinees has become a public health concern. To address the challenge, it is important to  investigate the prevalence of breakthrough infections by different SARS-CoV 2 variants, which will inform CoVID-19 vaccination and treatment.  

The authors conducted a clinical study and presented the data regarding the prevalence of breakthrough infection by SARS-CoV 2 variants in hospitalized vaccinees in Mexico. While the manuscript provides new information, its significance and impact are limited by the very small case numbers (n=53) in the study and the inclusion of partially vaccinated cases (n=7). The unvaccinated group had only 19 cases. In addition, as the investigation was performed only in one hospital in Mexico, it is unclear whether the data could represent the prevalence of breakthrough infection in Mexico as stated in the title. The authors concluded that the breakthrough infections that occurred in their hospital were mostly in patients vaccinated with CanSino. Is this data implying a low effectiveness of the vaccine against variant infections? Or it could be explained by the prevalence of the vaccine administered in the region? 

Author Response

Response to Reviewer 2 Comments

As new SARS-CoV 2 variants continue to emerge globally, breakthrough infection by variants in vaccinees has become a public health concern. To address the challenge, it is important to investigate the prevalence of breakthrough infections by different SARS-CoV 2 variants, which will inform CoVID-19 vaccination and treatment. The authors conducted a clinical study and presented the data regarding the prevalence of breakthrough infection by SARS-CoV 2 variants in hospitalized vaccinees in Mexico.

Point 1: While the manuscript provides new information, its significance and impact are limited by the very small case numbers (n=53) in the study and the inclusion of partially vaccinated cases (n=7). The unvaccinated group had only 19 cases.

Response 1: We thank the reviewer for their comments and revisions. Regarding this observation, it is indeed a limitation of the study, which we specified in the manuscript: “One limitation of this study is the inclusion of patients with partial vaccination, who only had one of the two doses recommended of Pfizer/BioNTech, AstraZeneca/Oxford, or CoronaVac [6, 16]. However, follow-up of instructions to complete vaccination scheme in people vaccinated with one dose is sometimes complicated. Another limitation is the small sample of patients included in the study. Better analyses could be performed if a higher number of samples can be studied.” However, the main aim of the study was to describe the prevalence of VOCs or VOIs in patients with partial or complete vaccination scheme in Monterrey, Mexico. This study allowed us to obtain a glimpse of the circulating variants in our hospital.

Point 2: In addition, as the investigation was performed only in one hospital in Mexico, it is unclear whether the data could represent the prevalence of breakthrough infection in Mexico as stated in the title.

Response 2: We thank the reviewer for this observation. We think this observation is correct, and thus we have modified the title to be more specific and avoid this confusion: “Prevalence of SARS-CoV-2 Variants of Concern and Variants of Interest in COVID-19 Breakthrough Infections in a hospital from Mexico”.  

Point 3: The authors concluded that the breakthrough infections that occurred in their hospital were mostly in patients vaccinated with CanSino. Is this data implying a low effectiveness of the vaccine against variant infections? Or it could be explained by the prevalence of the vaccine administered in the region?

Response 3: We thank the reviewer for this important observation. We have added this paragraph in the Discussion section: “Our results show higher prevalence of CanSino vaccine administration in our population (45.28%) compared to other vaccine brands in the same period. Furthermore, unlike other types of vaccines, patients vaccinated with CanSino and who had breakthrough infections, developed milder symptoms of COVID-19, which did not require hospitalization. Patients who had a complete vaccination scheme were less likely to develop a severe COVID-19 disease requiring hospitalization, compared to those who received a partial or incomplete immunization.” We also expanded the Abstract and Conclusion section.

Reviewer 3 Report

In this manuscript, the authors describe the prevalence of SARS-CoV-2 VOCs or VOIs in patients with partial or complete vaccination scheme in Monterrey, Mexico. However, current data are not enough to support their claims. Significant improvements are needed before consideration for publication.

  1. In abstract, the significance of this work and its enlightenments to other researchers should be provided.
  2. “BNT162b2 is a lipid nanoparticle-encapsulated nucleoside-modified RNA vaccine which encodes the receptor-binding domain (RBD) of the surface spike (S) glycoprotein of SARS-CoV-2”, the author’s statement is incorrect, the BNT162b2 encodes the full-length S protein of SARS-CoV-2.
  3. The table 1 contains 53 samples, which are not enough. Moreover, the analyzed data is related to one gender i-e Males, the author should provide data related to Females to make the data universal.
  4. In Figure 1, the yellow circles indicated the viruses isolated specifically in Mexico, however these viruses are not discussed in the manuscript. The authors are encouraged to discuss and explain the virus in the manuscript.
  5. “Current VOCs are Alpha (B.1.1.7), Beta (B.1.351, B.1.351.2, and B.1.351.3), Gamma (P.1, P.1.1, and P.1.2), and Delta (B.1.617.2, AY.1, AY.2, AY.3, and AY.4)”. However, the recently identified new variant named Omicron should also be included in the manuscript.
  6. The font size in the phylogenetic tree data in Figure 2 is too small to see clearly, and there is not sufficient information about the genomic sequence relationships. Moreover, some other data analysis can be added, for example, KEGG or GO analysis.
  7. The authors should revise the manuscript very carefully since many mistakes exist in the manuscript, e.g., Figure 1 on page 6 should be Figure 2; “After mRNA vaccines administration such as Pfizer or Moderna (mRNA-1273, Moderna, Moderna, Cambridge, MA)”; etc.
  8. Latest relative literatures are recommended to read: Chinese Chemical Letters, https://spj.sciencemag.org/research/2020/2402961/; Signal transduction and targeted therapy, https://doi.org/10.1038/s41392-021-00649-6; Exploration, https://doi.org/10.1002/EXP.20210051; Asian Journal of Pharmaceutical Sciences, https://doi.org/10.1016/j.ajps.2021.02.005.

Author Response

Response to Reviewer 3 Comments

In this manuscript, the authors describe the prevalence of SARS-CoV-2 VOCs or VOIs in patients with partial or complete vaccination scheme in Monterrey, Mexico. However, current data are not enough to support their claims. Significant improvements are needed before consideration for publication.

Point 1: In abstract, the significance of this work and its enlightenments to other researchers should be provided.

Response 1: We thank the reviewer for their comments and revisions. We have included the following statement in the Abstract: “SARS-CoV-2 breakthrough infections in our hospital occurred mostly in patients vaccinated with CanSino due to higher prevalence of CanSino vaccine administration in our population. These patients developed mild COVID-19 symptoms not requiring hospitalization. The significance of this study lies on the detection of SARS-CoV-2 variants compromising the efficacy of local immunization therapies in Monterrey, Mexico. CanSino immunization was efficient in decreasing hospitalization requirement in SARS-CoV-2 breakthrough infections.”

Point 2: “BNT162b2 is a lipid nanoparticle-encapsulated nucleoside-modified RNA vaccine which encodes the receptor-binding domain (RBD) of the surface spike (S) glycoprotein of SARS-CoV-2”, the author’s statement is incorrect, the BNT162b2 encodes the full-length S protein of SARS-CoV-2.

Response 2: We thank the reviewer for this observation. We have corrected this mistake within the manuscript.  

Point 3: The table 1 contains 53 samples, which are not enough. Moreover, the analyzed data is related to one gender i-e Males, the author should provide data related to Females to make the data universal.

Response 3: The table 1 refers to the total of samples included in the study, which was an observational case-control study of vaccine breakthrough infections in hospitalized patients in our unit, or who attended to the clinic, from April 19th, 2021, to August 11th, 2021. A total of 53 cases of COVID-19 vaccine breakthrough infections occurred in our hospital. Of those, vaccinated ambulatory patients were 30, and patients who required hospitalization were 23. Finally, 19 unvaccinated hospitalized patients (control group) were also included in the same period as the cases group. We have also added the data related to Females in Table 1.

Point 4: In Figure 1, the yellow circles indicated the viruses isolated specifically in Mexico, however these viruses are not discussed in the manuscript. The authors are encouraged to discuss and explain the virus in the manuscript.

Response 4: As the reviewer proposed, we included the description of yellow circles indicating Mexican isolates in the manuscript (lines 194 – 196, 198, 199, 202 – 204, 213, and 218 – 220).

Point 5: “Current VOCs are Alpha (B.1.1.7), Beta (B.1.351, B.1.351.2, and B.1.351.3), Gamma (P.1, P.1.1, and P.1.2), and Delta (B.1.617.2, AY.1, AY.2, AY.3, and AY.4)”. However, the recently identified new variant named Omicron should also be included in the manuscript.

Response 5: We thank the reviewer for this observation. We have added the Omicron variant and updated the VOI and VOC within the text. We also added this paragraph in the Results section: “At the time of the study, VOCs were Alpha (B.1.1.7), Beta (B.1.351, B.1.351.2, and B.1.351.3), Gamma (P.1, P.1.1, and P.1.2), and Delta (B.1.617.2, AY.1, AY.2, AY.3, and AY.4). VOIs were Eta (B.1.525), Iota (B.1.526), Kappa (B.1.617.1), Lambda (C.37), and Mu (B.1.621) [11]. Thus, the comparison studies were performed using this classification.”

Point 6: The font size in the phylogenetic tree data in Figure 2 is too small to see clearly, and there is not sufficient information about the genomic sequence relationships. Moreover, some other data analysis can be added, for example, KEGG or GO analysis.

Response 6: As the reviewer advised, we increased the font size in the phylogenetic tree data and now it is clearly visible. We also thank the reviewer for suggesting further analysis. However, after a careful examination, we consider that either a KEGG or a GO analysis will not provide additional crucial knowledge to the main aim of the study, which is to describe the prevalence of VOCs or VOIs in breakthrough infections in Monterrey, Mexico.

Point 7: The authors should revise the manuscript very carefully since many mistakes exist in the manuscript, e.g., Figure 1 on page 6 should be Figure 2; “After mRNA vaccines administration such as Pfizer or Moderna (mRNA-1273, Moderna, Moderna, Cambridge, MA)”; etc.

Response 7: We thank the reviewer for this observation. We have revised thoroughly within the manuscript and corrected all encountered mistakes.

Point 8: Latest relative literatures are recommended to read:

Chinese Chemical Letters, https://spj.sciencemag.org/research/2020/2402961/;

Signal transduction and targeted therapy, https://doi.org/10.1038/s41392-021-00649-6;

Exploration, https://doi.org/10.1002/EXP.20210051;

Asian Journal of Pharmaceutical Sciences, https://doi.org/10.1016/j.ajps.2021.02.005.

Response 8: We thank the reviewer for these suggested readings. However, as the manuscript is a brief report and the maximum of references allowed is 40, we cannot add more references.  

Round 2

Reviewer 2 Report

In the revised version, authors have addressed some of the critics raised by reviewers. The manuscript has been improved.  There are still questions need to be addressed.

  • Lines 27-29 in abstract: Due to very limited number of cases involved in the study and lack of detailed analysis of breakthrough infections in each vaccine group, it is inconclusive whether CanSino immunization was efficient in decreasing hospitalization requirement in SARS-CoV-2 breakthrough infections, as compared with other vaccines. It could be argued other factors might affect the results.
  • Line 320-321, regarding the prevalence of CanSino vaccine administration, authors stated ‘higher prevalence of CanSino vaccine administration in our population (45.28%) compared to other vaccine brands in the same period’. Apparently, the data (45.28%) represents the prevalence of Cansino vaccinee in the vaccinated patients involved in this study (Table 1). It needs to be revised by providing the prevalence of Cansino vaccination in the population in the region or Mexico.
  • Lines 321-323. ‘Furthermore, unlike other types of vaccines, patients vaccinated with CanSino and who had breakthrough infections, developed milder symptoms of COVID-19, which did not require hospitalization.’ This needs to be rewritten: some patients vaccinated with other vaccines also developed milder symptoms (Table 1), meanwhile some Cansino vaccinees developed severe diseases and had to be hospitalized.
  • Related to comment 3, in lines 336-337, authors concluded that ’Patients vaccinated with CanSino and who had breakthrough infections developed symptoms not requiring hospitalization.’  It is partially true.  The data clearly showed a more complicated picture. Some Cansino vaccinees developed severe diseases and had to be hospitalized.

Author Response

Point 1: Lines 27-29 in abstract: Due to very limited number of cases involved in the study and lack of detailed analysis of breakthrough infections in each vaccine group, it is inconclusive whether CanSino immunization was efficient in decreasing hospitalization requirement in SARS-CoV-2 breakthrough infections, as compared with other vaccines. It could be argued other factors might affect the results.

Response 1: We thank the reviewer for this important observation. We have deleted the sentence: “CanSino immunization was efficient in decreasing hospitalization requirement in SARS-CoV-2 breakthrough infections.” from the abstract.

Point 2: Line 320-321, regarding the prevalence of CanSino vaccine administration, authors stated ‘higher prevalence of CanSino vaccine administration in our population (45.28%) compared to other vaccine brands in the same period’. Apparently, the data (45.28%) represents the prevalence of Cansino vaccinee in the vaccinated patients involved in this study (Table 1). It needs to be revised by providing the prevalence of Cansino vaccination in the population in the region or Mexico.

Response 2: We thank the reviewer for this important observation. We have added the following suggested information: Our results show higher prevalence of CanSino vaccine administration in our population (45.28%) compared to other vaccine brands in the same period. Between April and May 2021, the national government implemented the administration of the CanSino vaccine exclusively in teaching staff, including the state of Nuevo Leon [41]. By the end of May, 87% of the teaching personnel in Mexico was fully vaccinated [42]. In addition, ours is a university teaching hospital, which attends a great part of the teaching staff.”

Point 3: Lines 321-323. ‘Furthermore, unlike other types of vaccines, patients vaccinated with CanSino and who had breakthrough infections, developed milder symptoms of COVID-19, which did not require hospitalization.’ This needs to be rewritten: some patients vaccinated with other vaccines also developed milder symptoms (Table 1), meanwhile some Cansino vaccinees developed severe diseases and had to be hospitalized.

Response 3: We thank the reviewer for this important observation. We modified the original paragraph to: “The majority of patients vaccinated with CanSino and who had breakthrough infections (66.66%), developed milder symptoms of COVID-19, which did not require hospitalization (p<0.01). Although our results were not conclusive regarding other types of vaccines, Pfizer/BioNTech and AstraZeneca/Oxford were 88.0% and 67.0% effective in reducing symptomatic Delta infection [19]. In our study, patients who had a complete vaccination scheme were less likely to develop a severe COVID-19 disease requiring hospitalization, compared to those who received a partial or incomplete immunization.”

Point 4: Related to comment 3, in lines 336-337, authors concluded that ’Patients vaccinated with CanSino and who had breakthrough infections developed symptoms not requiring hospitalization.’  It is partially true.  The data clearly showed a more complicated picture. Some Cansino vaccinees developed severe diseases and had to be hospitalized.

Response 4: We thank the reviewer for this important observation. We changed the sentence to: “In our population, most of the patients vaccinated with CanSino and who had break-through infections developed mild symptoms and few patients did require hospitalization.”